# Using Fitness Tracker Data to Overcome Pressure Insole Wear Time Challenges for Remote Musculoskeletal Monitoring

**DOI:** 10.3390/s24237717

**Published:** 2024-12-03

**Authors:** Cameron A. Nurse, Katherine M. Rodzak, Peter Volgyesi, Brian Noehren, Karl E. Zelik

**Affiliations:** 1Department of Mechanical Engineering, Vanderbilt University, Nashville, TN 37212, USA; 2Institute for Software Integrated Systems, Vanderbilt University, Nashville, TN 37212, USA; 3Department of Physical Therapy, University of Kentucky, Lexington, KY 40506, USA; 4Department of Biomedical Engineering, Vanderbilt University, Nashville, TN 37212, USA; 5Department of Physical Medicine and Rehabilitation, Vanderbilt University, Nashville, TN 37212, USA

**Keywords:** tibia shaft fracture, bone loading stimulus, telerehabilitation, wearable sensors, biomechanics

## Abstract

Tibia shaft fractures are common lower extremity fractures that can require surgery and rehabilitation. However, patient recovery is often poor, partly due to clinicians’ inability to monitor bone loading, which is critical to stimulating healing. We envision a future of patient care that includes at-home monitoring of tibia loading using pressure-sensing insoles. However, one issue is missing portions of daily loading due to limited insole wear time (e.g., not wearing shoes all day). Here, we introduce a method for overcoming this issue with a wrist-worn fitness tracker that can be worn all day. We developed a model to estimate tibia loading from fitness tracker data and evaluated its accuracy during 10-h remote data collections (*N* = 8). We found that a fitness tracker, with trained and calibrated models, could effectively supplement insole-based estimates of bone loading. Fitness tracker-based estimates of loading stimulus—the minute-by-minute weighted impulse of tibia loading—showed a strong fit relative to insole-based estimates (R^2^ = 0.74). However, insoles needed to be worn for a minimum amount of time for accurate estimates. We found daily loading stimulus errors less than 5% when insoles were worn at least 25% of the day. These findings suggest that a multi-sensor approach—where insoles are worn intermittently and a fitness tracker is worn continuously throughout the day—could be a viable strategy for long-term, remote monitoring of tibia loading in daily life.

## 1. Introduction

Tibia (shank bone) shaft fractures are one of the most common lower extremity fractures, and surgical fixation of the tibia is often necessary [1,2,3]. Surgical fixation surgeries have excellent results in terms of stabilizing the bone, enabling patients to start weight-bearing activities soon after surgery [4]. However, despite advances in surgical techniques [5,6], patient recovery 1 year following tibia shaft fixation surgery is often poor and inconsistent, with 65% of patients reporting an inability to perform pre-injury activities [7,8]. There is converging evidence that it is critical for the bone to experience enough loading to stimulate remodeling pathways after a fracture [9,10]. As such, a key role of clinicians after tibia fracture surgery is to develop and prescribe rehabilitation programs that progressively load the injured bone to stimulate recovery and restore functional abilities [11].

One limitation of the current rehabilitation programs stems from an inability to monitor how patients load their tibia in their daily lives. Patients are seen by clinicians during infrequent check-ups and consultations, leaving most of the patients’ daily activity—and associated bone loading—unseen and unknown by clinicians. This blind spot hampers a clinician’s ability to create and adapt effective rehabilitation programs and to understand other sources of bone loading in a patient’s daily life or whether patients are being too sedentary to stimulate recovery. There is a need for tools that offer healthcare professionals better insights into patients’ daily tibia bone loading, enabling personalized treatment and potentially fostering better care and recovery outcomes.

By integrating wearable sensors and software algorithms, we envision a future of patient care that includes at-home patient monitoring of bone loading. We have previously demonstrated that data from inertial measurement units (IMUs) and pressure insoles (termed *insoles*, for short) in shoes can be fused in biomechanically-informed machine learning algorithms to estimate tibia bone loading within 6% of traditional lab-based estimates [12,13]. Collectively, we refer to these as wearable sensor systems (hardware and software), and we believe they have the potential to revolutionize patient care by providing clinicians with continuous and personalized data on how patients are loading their tibia bone at home after surgery. However, there are still technical and practical challenges and scientific questions to answer before this vision can be realized.

One of the main challenges impeding the use of wearable sensor systems to remotely monitor bone loading in individuals over multiple days or weeks is the amount of daily sensor wear time. Musculoskeletal recovery and remodeling [14,15] are dependent on the volume and intensity of tissue loading. However, the amount of tissue loading measured depends on how much time the patient wears the sensors. While a typical adult may be awake for roughly 17 h per day [16], previous work observed that patients only wore insoles for between 4–5 h per day, in part because people do not wear shoes during every waking hour. Furthermore, the insole wear time varied widely day-to-day and between patients [17]. Thus, wearable sensor systems relying on in-shoe insoles are only expected to capture a portion of physical activity and bone loading each day.

There is a need to develop new methods for wearable sensor systems to obtain more complete, consistent, and comparable estimates of a patient’s bone loading across multiple days during remote monitoring, even if insole wear time varies day to day. Currently, limited or variable amounts of insole wear time can impede our ability to accurately monitor daily accumulated musculoskeletal loading. For instance, if a patient is physically active for 6 h but only wore insoles for 3 of those hours, then data from the insoles will underrepresent the loading experienced for that day. Now if that same patient were to perform the same amount of physical activity the following day but wore the insoles for the entire 6 h, then the insoles would report double the loading, even though the loading was the same on both days. This example illustrates a key technical gap when using insole-based sensor systems for remote patient monitoring: the inability to account for the variable wear time.

Fitness trackers are popular consumer devices that measure physical activity metrics, and while they cannot monitor bone loading, they have the potential for much longer wear times. Previous research demonstrates encouraging all-day compliance rates of 90% when participants were tasked with daily use of wrist-worn fitness trackers [18]. Fitness trackers use sensors such as inertial measurement units (IMUs), global positioning systems (GPS), optical heart rate sensors, and temperature sensors to monitor various physical activity metrics such as step count, heart rate, activity, and intensity. These metrics provide a high-level view of an individual’s activity and have been used to estimate physical activity intensity [19] and energy expenditure [20,21]. The high-level metrics from these devices could potentially be used as inputs to trained algorithms to extrapolate tibia bone loading during periods when patients are not wearing insoles. However, we do not yet know how to leverage fitness trackers (e.g., wrist-worn, arm-worn, ring-style devices) to complement remote musculoskeletal load monitoring derived from other body-worn sensors (e.g., insoles) or if fitness trackers can help overcome the patient wear time issues with insoles.

The objective of this study was two-fold. First, we sought to develop a model to estimate tibia bone loading solely from fitness tracker inputs to address the technical gap related to inconsistent insole wear time and to provide a novel way to fill gaps in insole data during remote collections. Second, we aimed to characterize how errors in daily tibia load estimates increase with less insole wear time. The second objective helps inform whether there is a minimum amount of time a person needs to wear insoles each day for effective remote monitoring of tibia bone loading.

## 2. Materials and Methods

### 2.1. Summary

First, we synchronously collected data from in-shoe insoles and a wrist-worn fitness tracker on 8 participants in their daily life. Second, we estimated time-series tibia bone loading (i.e., force) using the insole data and previously published methods.

Third, to address objective 1, we trained a model to estimate tibia loading using only fitness tracker metrics. The specific output from this fitness tracker model was a metric called loading stimulus (LS), which represents the cumulative loading experienced by the tibia bone on a minute-by-minute basis. We then compared the LS estimates from the fitness tracker model to the LS estimates computed from the insole data to evaluate accuracy of our trained model.

We were unable to achieve accurate LS estimates using a generic (participant-independent) model and fitness tracker data. We, therefore, created a more advanced model with an additional participant-specific calibration to reduce LS errors and variability. Throughout the manuscript, we refer to these models and their LS outputs as *generic* and *calibrated*, respectively.

Fourth, to address objective 2, we built a data-driven simulation for each participant to explore how insole wear time affects the accuracy of daily tibia LS estimates. We summed LS estimates over 9–10 h per participant to compute a daily load stimulus (DLS) summary metric. We initially used 100% insole data to estimate LS and DLS and to represent the best-case scenario of someone wearing their insoles all day. Next, we ran simulations with varying amounts of insole data (e.g., 90% insole data, then 80%, etc.) to represent different insole wear time scenarios. We used the calibrated fitness tracker model to fill gaps in the LS estimates. We then characterized how DLS estimates changed with decreasing insole wear time, relative to the best-case scenario.

### 2.2. Data Collection

We recruited a convenience sample of 8 participants (4 male, 4 female, aged 21–29 years) to take part in this study, and they provided informed written consent following ethical approval from the Institutional Review Board at Vanderbilt University. We have previously used a similar sample size to develop and demonstrate feasibility of wearable sensor algorithms for musculoskeletal monitoring [12,13,22,23]. We equipped participants with a properly sized Moticon OpenGo insole (Munich Germany Figure 1) in each shoe and a Garmin Vivosmart 5 fitness tracker (Olathe, KS, USA Figure 1) worn on one wrist. Participants wore these sensors for 9–10 h in their daily life and we instructed participants to go about their normal daily activities and keep their shoes on as much as practical. Participants were also asked to perform a series of exercises in the lab (for about 20 min), once in the morning at the start of the data collection and once again in the afternoon. These exercises were representative of rehabilitation exercises commonly performed during tibia fracture recovery (see Appendix A for detailed description). The insoles continuously collected in-shoe pressure data and 6-axis acceleration and gyroscope data from an IMU inside each insole at 25 Hz. The fitness tracker contains several sensors but only exports certain metrics (detailed below) at 0.017 Hz (once per minute).

### 2.3. Calculating Tibia Force from the Insole Data

We calculated time-series tibia compression force ($F_{Tibia}\left(t\right),$ also termed *loading*) of the right leg using the total force (F(t)) and center of pressure (CoP(t)) from the insoles and previously established physics-based methods [12,24] (Equation (1)). This insole-based estimate of tibia force was used to compute our target LS for all subsequent processes.
(1)$$F_{Tibia}\left(t\right)=F\left(t\right)+\frac{\left(CoP\left(t\right)-x\right)\cdot F\left(t\right)}{r}$$
where r is the Achilles tendon moment arm relative to the ankle joint center, assumed to be a constant 5 cm [12], and x is the horizontal distance that we measured from the back of the insole to each participant’s ankle joint center when the shoe was flat on the ground [24].

### 2.4. Objective 1: Developing a Model to Estimate Tibia Load from Fitness Tracker Data

#### 2.4.1. Model Inputs

Model inputs were exported from the fitness tracker. These metrics included activity time, step count, scaled steps, activity level, and scaled activity. Each metric was extracted on a 1-min interval using Labfront software (version 1.0.52). These fitness tracker metrics were chosen due to their general relevance to movements associated with tibia loading during locomotion. Scaled features involved multiplying the selected signal (i.e., step count or activity time) by the activity level measured by the fitness tracker during the same minute interval. In the Labfront software, activity level was calculated using a proportional integrating method, a measure of activity magnitude or motion vigor, defined as the area under the curve of the accelerometry signal [25,26]. Activity time is computed as the duration in which the absolute value of the acceleration magnitude from the fitness tracker exceeded 50 mG.

#### 2.4.2. Model Target

We defined LS, a target bone load metric for the model to estimate. LS represents the mechanical stimulus experienced by the tibia over 1 min. This time interval was selected because the fitness tracker we used only exports metrics once per minute. This exponential relationship was used to reflect how loading stimulates bone healing processes (e.g., remodeling), using m = 4 as the exponent [27] (Equation (2)). There is evidence that load-induced tissue damage stimulates biological remodeling [27,28,29,30], and that linearly summing loads over time does not accurately reflect the damage experienced [31]. Thus, LS is non-linearly related to tibia force in Equation (2) [31,32]:(2)$$LS={\left(\int_{t_{a}}^{t_{b}}F_{Tibia}\right)}^{m}$$
where *t_a_* and *t_b_* are 1 min apart.

#### 2.4.3. Generic Model Development

We initially developed a generic model to estimate LS from the fitness tracker metrics. We selected least absolute shrinkage and selection operator (LASSO) regression as a suitable approach for developing our model based on prior success using this type of algorithm to estimate tibia bone loading from discrete metrics [12,13]. The LASSO regression is a technique for statistical modeling and machine learning that uses a least squares model with L1 regularization. This regularization penalizes the sum of the absolute values of the coefficients and forces a subset of learned coefficient weights to zero, helping to build more interpretable models and prevent overfitting [33]. The LASSO model was trained using k-fold cross validation by participant, which means the data from seven participants were used to train the model. The model was then applied to the single participant excluded from the training set. This process was repeated for each participant [34], resulting in 8 different regressions representing a generic model without any prior knowledge of the individual user.

#### 2.4.4. Participant-Specific Calibration Model

We performed an additional calibration step to better estimate LS from fitness tracker metrics. We created a second LASSO model for each participant. The inputs to this calibration model were the LS estimates from the first LASSO model, as well as the fitness tracker metrics. The target was the LS estimate computed from the insole.

The LASSO calibration model was trained using a cross validation by fold, using a quarter of the data to train a model, and then applied to the rest of the data from that same participant (Figure 1). This represents a scenario where a user wore both their insoles and fitness tracker for 25% of their day, but then for the remaining 75% of the day, they only wore the fitness tracker. See Section 2.5 for a deeper analysis of insole wear time effects (i.e., what happens when different percentages of insole data were used for training this calibration model).

#### 2.4.5. Evaluation

We calculated the coefficient of determination (R^2^) to evaluate our models and quantify the goodness of fit of each model output to the target LS. R^2^ was calculated between insole estimates of LS (target) and those outputted by the generic and calibrated fitness tracker models for each participant. If R^2^ is negative, it means that the model fit is worse than a horizontal line that consistently estimates the average of the target. If the R^2^ is equal to 1, it means that the model output perfectly predicts the target metric.

### 2.5. Objective 2: Simulating How the Amount of Insole Data per Day Affects Daily Loading Estimates

We built a data driven simulation that estimates daily LS (i.e., cumulative LS over an entire day). We explored scenarios where insoles were worn only part of the day and the remaining LS estimates were computed using the fitness tracker data with the calibrated model, per Section 2.4. LS estimates from periods with insole data were used to train participant-specific calibration models. These calibration models were then applied to intervals without insole data to estimate LS using the fitness tracker only (Figure 2). Finally, we calculated how the percentage of insole data collected each day influenced the daily LS estimated, relative to the best-case scenario of having 100% of the insole data to compute LS.

#### 2.5.1. Daily Loading Stimulus (DLS)

Our long-term goal is to develop and validate a new capability that enables remote monitoring of individuals after tibia shaft fracture surgery (or potentially other lower-limb injuries). We seek to understand how tibia bone loads experienced by patients in daily life influence recovery and how bone loading changes over weeks and months. When looking across multiple days, it is useful to define a cumulative summary metric, which we term daily load stimulus (DLS). Daily load stimulus is an empirically-derived expression relating tissue damage accumulation and the mechanical stimulus for bone remodeling [28,29,30,35] (Equations (3) and (4)). For different movement tasks (e.g., walking, running, jumping), the stimulation induced by each task (j) is related to the stress magnitude $\sigma_{j}$ and the number of loading cycles $n_{j}$ (e.g., steps). In some formulations, $\sigma_{j}$ represents the peak stress for a given task (Equation (3)) [27,29,30,35], while in other formulations, it represents the time-series or time-integrated stress (i.e., impulse, Equation (4)) [31,32]. Since localized stress is generally impractical to measure in vivo, indicators of stress such as tissue force or biomechanical moments [12,13,22] are often used as inputs to these expressions. The exponent *m* in Equations (2)–(4) reflects the relative contribution of stress magnitude based on animal and cadaver studies and has typically been assigned a value of 4 [27,28,29,30].
(3)$$DLS={\left[\sum_{j=1}^{k}n_{j}{\left(\sigma_{j}\right)}^{m}\right]}^{\frac{1}{2m}}$$
(4)$$DLS={\left[\sum_{j=1}^{k}n_{j}\int_{t_{i}}^{t_{f}}{\left(\sigma_{j}\left(t\right)\right)}^{m}dt\right]}^{\frac{1}{2m}}$$

In our analysis, we used the time-integral method (Equation (4)), where every minute of data was treated as a unique, discrete task (j), such that $n_{j}$ was always 1. Furthermore, $\sigma_{j}(t)$ was approximated using $F_{Tibia}(t)$ such that $\int_{t_{i}}^{t_{f}}{\left(\sigma_{j}(t)\right)}^{m}dt$ was equivalent to ${LS}_{j}$, as defined in Equation (2), thus yielding Equation (5) for DLS:(5)$$DLS={\left[\sum_{j=1}^{k}{LS}_{j}\right]}^{\frac{1}{2m}}$$

#### 2.5.2. Simulated Days

We simulated 300–400 days for each participant using the empirical data we collected. Each simulated day consisted of the total duration of data collected for each participant in the study (i.e., 9–10 h). Simulated days varied in the percentage of insole data used and, therefore, the percentage of the day that needed to be filled with fitness tracker estimates of LS. We used an incremental sliding window approach, where the time interval captured inside the window represented the duration of time that the participant wore the insoles on that day. We started with a 30-min window size (i.e., 5% insole data for a 10-h day), and once a window of a given size moved across the entire day, the size was increased by 5-min. For each simulated day, a participant-specific calibration model was created based on the insole and fitness tracker data within the sliding window, following the method outlined in Section 2.4.4. This calibration model was then applied to the fitness tracker data outside the sliding window, and the process was repeated for each simulated day.

The DLS from each simulated day was calculated by concatenating the insole LS estimates from inside the sliding window and the fitness tracker LS estimates from the rest of the day (Figure 2), then applying Equation (5).

#### 2.5.3. Evaluation

The mean absolute percent error (MAPE) for each simulated day was obtained by comparing the combined fitness tracker and insole DLS to the DLS estimated from only insole data. To summarize the results, the simulated days were categorized into bins based on the percentage of available insole data, with each bin representing a 5% of the day increment (e.g., 0–5%, 5–10%, 10–15%, and so on). For each participant, the average MAPE was calculated within each bin. We then averaged these values across participants to obtain the overall mean and standard deviation for each bin.

## 3. Results

Objective 1: The LS estimates from the generic model showed a poor fit to the target, with an average R^2^ of −1.75 (Figure 3, Table 1), signifying that, on average, this model fit was worse than simply estimating the average target LS. In contrast, the calibrated LS estimates demonstrated a strong fit, achieving an average R^2^ of 0.76 (Figure 3, Table 1).

Objective 2: The average error in DLS estimates using only fitness tracker data was 12.6% and varied between about 5–23% across individuals (Figure 4). This error increased until insole data were used from more than about 5% of the day (about 30 min), at which point, the error began to decrease following a pattern similar to exponential decay. The average error in DLS estimates decreased below 5% when insole data from 25% of the day (about 2.5 h) were included.

## 4. Discussion

We developed a model that estimates tibia bone loading stimulus (LS) directly from fitness tracker metrics to support longitudinal, remote monitoring and help overcome insole wear time challenges. We found that a generic fitness tracker model did not provide accurate estimates of LS (R^2^ = −1.75), indicating that the fitness tracker was not able to fully replace the need for insoles when monitoring tibia bone loading. However, by adding an easy-to-implement, participant-specific calibration, we greatly improved the accuracy of LS estimates from fitness tracker data (R^2^ = 0.76). This result indicates that a fitness tracker can be used to supplement and gap-fill insole-based estimates of LS, which is useful when insoles are only worn for a portion of the day. We also characterized how daily loading stimulus (DLS) estimation accuracy changes with reduced insole wear time. Our results indicated that wearing insoles 25% of the day (e.g., 2.5 out of 10 h) was sufficient, with the calibrated fitness tracker model, to achieve an average DLS error of under 5%. These findings suggest that a multi-sensor approach—where insoles are worn intermittently, and a fitness tracker is worn continuously throughout the day—could be a viable strategy for long-term, remote monitoring of tibia loading in daily life. Below, we discuss the key considerations, potential applications, and limitations of implementing this approach in real-world practices.

### 4.1. Individual Variability and the Benefits of Participant-Specific Calibration

Individual behaviors pose a challenge for developing a generic model that uses fitness tracker data to estimate tibia bone loading across different users. We observed substantial variability in fitness tracker metrics between participants during periods of both high and low tibia loading. For instance, participants with smaller arm swings during walking recorded lower activity levels than those with larger arm swings, even when tibia loading was comparable. Similarly, individuals who used a lot of expressive hand gestures during sedentary activities resulted in inflated activity levels. The type and frequency of hand gesturing not only varies between individuals but is also well-known to vary widely across different countries and cultures. Collectively, this behavioral variability complicates the ability to identify tibia loading solely from wrist-worn fitness trackers through generalized methods.

We found participant-specific calibrations to be a practical and effective way to improve tibia loading estimates from fitness tracker data. The key was capturing enough simultaneous data—from both the insoles and the fitness tracker—for each individual that we could model and account for their unique movement patterns (e.g., arm swing magnitude, hand gesture frequency). Importantly, these data can be captured easily during a person’s normal daily activities and without the need for any special tasks or instructions. Without participant-specific calibration, insole data from about 60% of the day were needed to achieve less than 5% error in DLS (Figure 5). However, with the calibration model, this decreased to only needing insole data from 25% of the day (Figure 5). For a 16-hour day (awake time), this is the difference between participants needing to wear insoles (and shoes) for almost 10 h vs. only 4 h. The former (10 h) is expected to be prohibitive for many users and use cases, whereas the latter (4 h) seems achievable, and it aligns with insole wear time by patients in a previous remote monitoring study [17]. We also observed that if calibrations were performed with very small amounts of insole data (less than one hour), there was not enough loading information to build a robust model, resulting in increased error (Figure 5). Thus, participant-specific calibrations offer a promising and practical solution for full-day tibia load monitoring in daily life, but it will be critical that users wear the insoles for enough time to build accurate calibration models for the supplementary fitness tracker.

### 4.2. Impacts on Clinical Patient Monitoring

This new approach combines wearable sensors (insoles and fitness trackers) and trained models to obtain more complete daily estimates of bone loading. As discussed in the Introduction, we believe these more complete estimates are critical to obtaining comparable bone loading estimates across multiple days or weeks. This capability could unlock new possibilities for both clinical practice and longitudinal research.

This approach will allow researchers to track tibia loading across extended periods, for instance, throughout recovery from tibia fracture surgery. Previous work has observed 4–5 h of daily insole wear time, which corresponds to roughly 25–30% of a 16-h day [17]. With the calibrated model developed in this study, this amount of insole wear time each day would be expected to result in less than 5% error in DLS estimates (Figure 5). By consistently monitoring daily bone loading over multiple months during recovery from fracture, researchers may be able to identify patterns and establish benchmarks for the amount of tibia loading that leads to the most favorable patient outcomes.

In the future, we envision that clinicians using this approach may be able to make more targeted adjustments to rehabilitation plans and optimize exercise regimens or activity restrictions. For instance, rehabilitation could be personalized and informed by real-world data rather than by infrequent clinic-based evaluations, generic guidelines, or patient self-reports (which can sometimes be unreliable). The LS estimation model (Figure 2) developed in this study could provide clinicians with a more comprehensive view of a patient’s bone loading patterns during everyday life. For example, if a patient consistently underloads their tibia during daily activities, clinicians can provide guidance to increase activity safely while avoiding excessive loading that might cause setbacks in recovery. This remote monitoring approach represents a significant step toward more personalized and data-driven rehabilitation programs.

### 4.3. Other Monitoring Applications

This study focused on monitoring tibia loading that stimulates bone remodeling after fracture surgery, but the framework could also be adapted for other applications or musculoskeletal tissues. For example, a number of different insoles have been validated to estimate ground reaction forces [36,37] or to estimate other musculoskeletal loading metrics [12,13,22,23,24].

The same wearable sensors and similarly trained models used in this study could be leveraged to estimate cumulative damage metrics that have been used in ergonomic and sports applications to relate musculoskeletal tissue loading to injury risk [38,39,40]. Previous research has demonstrated that pressure insoles and IMUs can estimate loading, damage, and injury risk relevant to running and ergonomic contexts [12,13,22,23]. However, the real-world applications would likely have analogous issues related to limited wear time of insoles. Fitness trackers are currently more transparent and easier to integrate into daily life than insoles and could help facilitate longer term data collections. The fitness tracker gap-filling models presented herein have potential for broad applicability to any insole-based, shoe-worn, or body-worn monitoring that requires full-day or multi-day use but where daily wear time of the primary sensor system is limited.

### 4.4. Limitations

There are important limitations with the current study, as well as opportunities for future research. First, we collected a convenience sample of 8 participants, consistent with prior feasibility studies and engineering/algorithm development studies using wearable sensors to estimate musculoskeletal loads [12,13,22,23,24]. For example, Elstub et al. [13] developed a machine learning model to estimate peak tibial bone force during running with wearable sensor data from 9 participants. It is challenging to select a sample size for new engineering development projects (e.g., developing new estimation models involving wearable sensors and machine learning, then characterizing how they perform with different data inputs). Traditional sample size and power analysis approaches generally are not applicable to this type of research because the analysis methods are not (and cannot be) fully known a priori. Rather, to develop new estimation methods, it is typically necessary to start with a modest sample size and dataset, then demonstrate initial feasibility and characterize accuracy, as in this study. Fortunately, cross-validation techniques helped provide confidence in data interpretation and allowed us to gain some insight into model generalizability, even with smaller sample sizes. Similar results observed across individuals with the calibrated LS model (Table 1) suggest that increasing the sample size is unlikely to alter the primary takeaways of this study. Additionally, while our study included 8 individuals, we collected over 9 h of uncontrolled real-world activity data per participant, representing a broad range of conditions. This dataset is more representative than what is typically gathered in controlled lab studies, where monitoring is often limited to an hour or two of data from a restricted subset of tasks. Ultimately, the initial algorithm development work and accuracy characterizations detailed in this manuscript were necessary first steps to complete and these provide the critical scientific foundation for subsequent research studies. Future studies should investigate larger sample sizes and longer time durations (e.g., multi-day or multi-week) to further explore generalizability and accuracy and to further refine LS estimation models.

Second, our participants were all healthy individuals within a similar age range (21–29 years). In clinical populations, factors such as age, physical fitness, immobilizing boots, and altered movement patterns introduce additional complexity. We believe that the development of participant-specific calibration models should also work for various patient groups, but further research and validation are warranted for clinical use cases.

Third, we only collected a single day of data for each participant, and we have not yet assessed the repeatability of our method over multiple days. Clinical interventions for tibia shaft fractures often span several months, during which patient activity levels and recovery trajectories can vary significantly. Evaluating the day-to-day repeatability of these sensors and models is, therefore, essential to ensure their reliability in capturing consistent and representative loading patterns over extended periods. Future research should focus on multi-day or longitudinal data collections to better understand intra-individual variability and to assess the reliability of the LS estimates over time.

Fourth, we boiled our analysis down to the percentage of the day that insoles are worn. However, it is actually more complicated and there are alternative ways to define minimum insole wear time guidelines. To expound, our simulations indicated that, on average, 2.5 h of insole data (per 10-h day) were sufficient to reduce DLS error below 5%. However, the DLS accuracy was more dependent on the type of activity during the hours when insoles were worn than the duration itself. DLS errors from 2.5 h of insole data ranged from 1.2% to 37.1% across simulated days. For instance, if participants wore the insoles while sitting for 2.5 h, the resulting calibration model would be inadequate for estimating loading over the rest of the day. From our simulation, 2.5 h on average contained enough physical activity and loading to build effective calibrations without specific instructions to participants. However, future studies should further assess the repeatability of this method, ideally in the target clinical population, and there may be other practical ways to obtain daily calibration data from the insoles. For instance, we might discover that patients only need to wear the insoles during a few short bouts of physical activity each day. Finally, we only presented results for the right insole. However, we repeated the analysis methods for the left insole and obtained similar results. These were omitted for brevity and because they did not add to or alter any of the study conclusions.

## 5. Conclusions

We found that a wrist-worn fitness tracker coupled with trained models could effectively and practically supplement remote monitoring of tibia bone loading with pressure sensing insoles, helping to overcome insole wear time challenges.

## Figures and Tables

**Figure 1 sensors-24-07717-f001:**
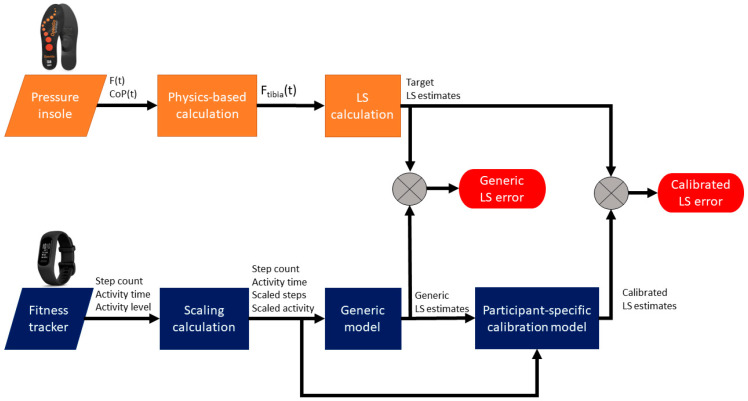
Signal flow chart for LS estimation models. This shows how signals were processed to compute tibia loading stimulus (LS) estimates from the fitness tracker data. Orange represents data/estimates from the insole and blue represents data/estimates from the fitness tracker.

**Figure 2 sensors-24-07717-f002:**
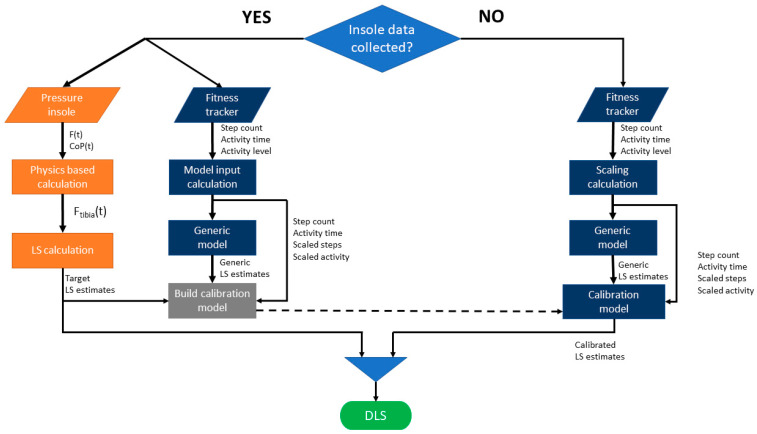
Signal flow diagram for simulated days. When insole data are collected, both insole and fitness tracker data are used to build a participant-specific calibration model for that simulated day. When insole data are not available, fitness tracker inputs are processed through the generic model first, then the calibration model that was built is applied (dashed line) to estimate the calibrated loading stimulus (LS). The daily loading stimulus (DLS) is calculated by combining LS estimates from the insole, when available, with LS estimates from the fitness tracker when insole data are absent, then applying Equation (5).

**Figure 3 sensors-24-07717-f003:**
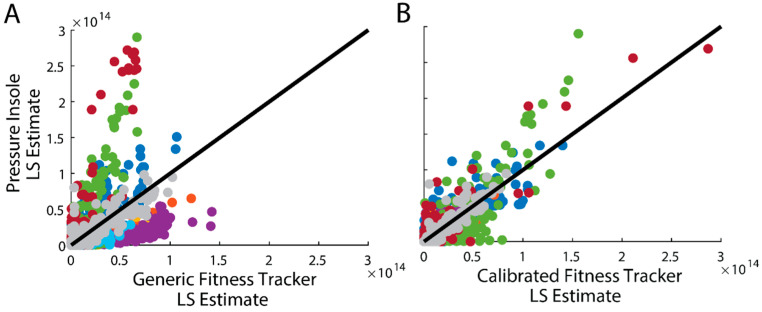
Loading stimulus (LS) computed from the insole vs. (**A**) generic and (**B**) calibrated fitness tracker LS estimates. Each color represents a different participant, and each data point a different minute of the data collection. The black line represents the perfect relationship between fitness tracker and insole LS estimates.

**Figure 4 sensors-24-07717-f004:**
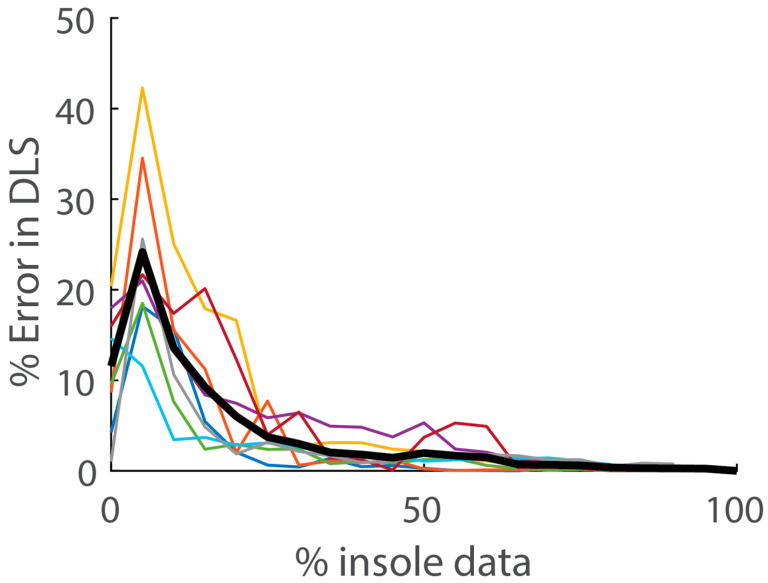
The percentage of insole data used vs. the error in DLS estimates during simulated days. Errors in DLS are computed relative to the best-case scenario, in which insoles are worn 100% of the day. Each thin colored line is the average for an individual participant. The thicker black line represents the inter-participant average.

**Figure 5 sensors-24-07717-f005:**
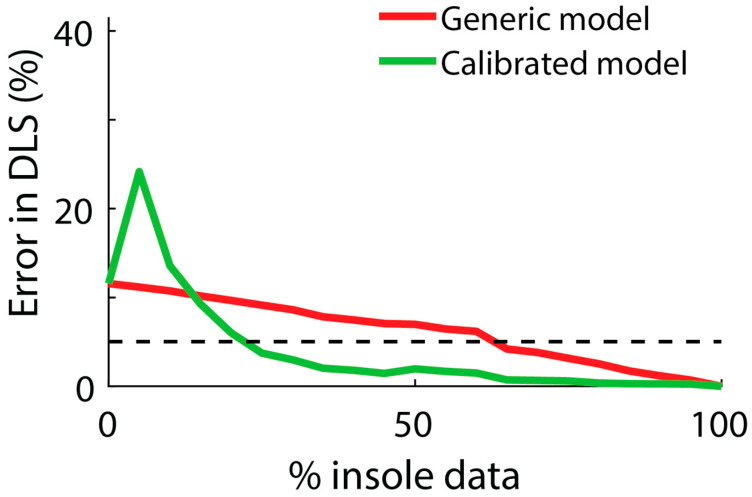
The percentage of insole data used vs. the average error in DLS estimates during simulated days when applying the generic model vs. the calibrated model. Errors in DLS are computed relative to the best-case scenario in which insoles are worn 100% of the day. Each line represents the average across all participants. A dashed horizontal black line representing 5% error in DLS is shown for visual reference.

**Table 1 sensors-24-07717-t001:** Coefficient of determination (R^2^) for generic and calibrated fitness tracker estimates of LS relative to insole estimates of LS.

	Coefficient of Determination	
Participant	Generic	Calibrated
1	0.72	0.80
2	−0.15	0.86
3	−6.51	0.67
4	−6.07	0.65
5	0.40	0.72
6	−3.51	0.67
7	0.34	0.85
8	0.79	0.77
Average	−1.75 ± 0.71	0.74 ± 0.06

## Data Availability

The data are available upon request.

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
