# Peer review of "Using Fitness Tracker Data to Overcome Pressure Insole Wear Time Challenges for Remote Musculoskeletal Monitoring"

_sensors, 2024, doi:10.3390/s24237717_

Round 1

Reviewer 1 Report

Comments and Suggestions for Authors

The tibial load monitoring method based on wrist fitness trackers and insoles proposed in this article has certain innovation and practical application prospects, solving the problem of data loss caused by wearing time limits in existing monitoring methods. The research design is reasonable, and the experimental results show good model fitting effect. However, the sample size and validation time are relatively short, and the adaptability and long-term stability of the model still need further validation. Suggestions for further improvement in sample size, error analysis, and application convenience, such as: 1. Increasing sample size and representativeness: This study only collected data from 8 participants, which may limit the generalizability and generalizability of the research results. To improve the representativeness of the research and the broad applicability of the results, it is recommended to expand the sample size in subsequent studies and include participants of different ages, genders, and health conditions; 2. Extended duration of data collection: This study only collected data for one day for each participant, which may not fully reflect the variability in individual daily activities. It is recommended to extend the data collection time to capture a wider range of behavioral patterns and evaluate the repeatability and stability of the model on different days, thereby enhancing the representativeness and reliability of the study.

Reviewer 2 Report

Comments and Suggestions for Authors

INTRODUCTION

While the introduction mentions the problem of inconsistent rehabilitation monitoring and the limitations of insole-based systems, it does not clearly define the specific gap in the existing literature or clinical practice that this study addresses. For example: How does this study differ significantly from prior research utilizing fitness trackers or insoles? What makes it novel?

METHODS

A sample size of 8 participants is very small, especially for a study aiming to develop a model or validate a monitoring method. While prior use of a similar sample size is mentioned as justification, no rationale is provided for why this size is sufficient for the current study.

The description of the lab exercises is vague ("representative of rehabilitation exercises") and does not specify: The types or intensity of exercises performed; How these exercises align with rehabilitation protocols for tibia fracture recovery.

RESULTS

This is a well written section with high quality and informative tables and figures.

DISCUSSION

Line 433: Reader would benefit from being introduced to other insoles that have been validated to measure load-related metrics during multiple functional activities, such as Loadsol (https://pubmed.ncbi.nlm.nih.gov/30427263/) and XSENSOR (https://pubmed.ncbi.nlm.nih.gov/37696840/)
